# Species and condition shape the mutational spectrum in experimentally evolved biofilms

Guohai Hu,[1,2,3] Yue Wang,[1,2,4] Xin Liu,[1,2,4] Mikael Lenz Strube,[5] Bo Wang,[1,2,6] Ákos T. Kovács[3,7]

**ABSTRACT** Laboratory experimental evolution provides a powerful tool for studying microbial adaptation to different environments. To understand the differences and similarities of the dynamic evolutionary landscapes of two model species from the *Bacillus* genus as they adapt to abiotic and biotic surfaces, we revived the archived population samples from our four previous experimental evolution studies and performed longitudinal whole-population genome sequencing. Surprisingly, higher number of mutations, higher genotypic diversity, and higher evolvability were detected in the biotic conditions with smaller population size. Different adaptation strategies were observed in different environments within each species, with more diversified mutational spectrum detected in biotic conditions. The insertion sequences of *Bacillus thuringiensis* are critical for its adaptation to the plastic bead-attached biofilm environment, but insertion sequence mobility was a general phenomenon in this species independent of the selection condition. Additionally, certain parallel evolution has been observed across species and environments, particularly when two species adapt to the same environment at the same time. Furthermore, our results suggest that the population size might be an important driver of evolution. Together, these results provide the first comprehensive mutational landscape of two bacterial species' biofilms that is adapted to an abiotic and biotic surface.

**IMPORTANCE** Biofilm formation is a vital factor for the survival and adaptation of bacteria in diverse environmental niches. Experimental evolution combined with the advancement of whole-population genome sequencing provides us a powerful tool to understand the genomic dynamic of evolutionary adaptation to different environments, such as during biofilm development. Previous studies described the genetic and phenotypic changes of selected clones from experimentally evolved *Bacillus thuringiensis* and *Bacillus subtilis* that were adapted under abiotic and biotic biofilm conditions. However, the full understanding of the dynamic evolutionary landscapes was lacking. Furthermore, the differences and similarities of adaptive mechanisms in *B. thuringiensis* and *B. subtilis* were not identified. To overcome these limitations, we performed longitudinal whole-population genome sequencing to study the underlying genetic dynamics at high resolution. Our study provides the first comprehensive mutational landscape of two bacterial species' biofilms that is adapted to an abiotic and biotic surface.

**KEYWORDS** experimental evolution, parallelism, population size, *Bacillus subtilis*

B iofilms are matrix-enclosed microbial communities that adhere to biotic or abiotic surfaces. These complex assemblages confer emergent properties to their inhabitants and represent a much higher level of organization than free-living bacterial cells do (1, 2). The biofilm lifestyle not only protects bacteria to survive in harsh environments but also facilitates the colonization of new niches during dispersal from these microbial

Address correspondence to Bo Wang, wangbo@cngb.org, or Ákos T. Kovács, a.t.kovacs@biology.leidenuniv.nl.

Guohai Hu and Yue Wang contributed equally to this article. Co-first authors Hu and Wang are in the order based on their contribution to writing.

The authors declare no conflict of interest.

See the funding table on p. 17.

clusters. Bacteria form biofilms on almost all natural and artificial surfaces (2, 3), utilizing diverse mechanisms that depend on environmental conditions and specific species (4). Biofilms can be categorized into three models depending on the type of occupied niche, including pellicle biofilms at the air-liquid interface, colony biofilms at the air-solid interface, and submerged biofilms at the solid-liquid interface (5, 6). Regardless of the type, these biofilm types can be formed under either abiotic conditions or in connection with a host. Members of the Gram-positive *Bacillus* genus form various types of biofilms exhibiting either beneficial or pathogenic impact. Here, we focused on two model species from the *Bacillus* genus and experimental evolution of their biofilms.

*Bacillus thuringiensis* is commonly used as a biological pesticide belonging to the *Bacillus cereus sensu lato* group, and its spores can be isolated from diverse environments (7). *Bacillus subtilis* is a soil-dwelling, non-pathogenic bacterium that is commonly found in association with plants and their rhizosphere (6). Both species can form biofilms in diverse environments. They share a large number of transcriptional factors, including Spo0A, $\sigma^B$, SinI, and SinR (8), which play crucial roles in the intertwined regulation of sporulation and biofilm formation (9–13). Between these two species, however, there are also some important differences in the regulatory pathways and transposable elements. For example, SigD, the motility-specific sigma factor, and the DegU/DegS two-component system, are absent from the species of the *B. cereus sensu lato* group; while the virulence regulator PlcR, which plays an important role in *B. cereus* and *B. thuringiensis* physiology (14, 15), is absent in *B. subtilis*. Besides, diverse insertion sequences (ISs) have been described in different *B. thuringiensis* isolates (16–19), while only limited number of ISs have been reported in *B. subtilis* (20). ISs are the simplest transposable elements and play important role in shaping their host genomes (21). Insertions of IS elements can result in both gene inactivation and activation, or alter the expression of neighboring genes (22). In addition, IS-mediated changes have been described to both promote and constrain evolvability of *Escherichia coli* in a long-term evolution experiment (LTEE) (23).

Experimental evolution has been widely used to study the evolutionary processes that occur in experimental populations in response to the surrounding environment (24–26). Experimental evolution provides a powerful tool to study microbial adaptation to different environments in real time. Combined with the advancement of whole-population genome sequencing, it is possible to understand the genomic dynamic of evolutionary adaptation (27–30). Plenty of evolutionary studies have been performed on planktonic forms of bacteria and yeasts, under diverse environmental conditions (27, 30–35). In contrast, only a relatively limited number of evolution experiments have been performed with biofilm populations (28, 36–40). Most of the biofilm evolution studies have focused on the quick emergence of morphotypic, phenotypic, and genotypic variation within biofilms. In biofilms, astonishing parallelism has been observed at the different biological hierarchy levels, from fitness level to gene level and even nucleotide level, both between replicate lineages within the same evolution experiment and across different evolution experiments (24).

Previously, we have experimentally evolved *B. thuringiensis* and *B. subtilis* using generally comparable adaptation concepts. For *B. thuringiensis,* experimental evolution was performed using two approaches, a plastic bead-attached biofilm model (Bth_bead) (41) and *Arabidopsis thaliana* root colonization model (Bth_root) (42). Similarly, *B. subtilis,* a static air-medium floating biofilm transfer mode (Bs_pellicle) (43), and an *A. thaliana* root colonization model (Bs_root) (44) were used. Thus, for each bacterial species, biofilm formation proceeded under abiotic conditions (plastic bead or air-liquid interface) and on biotic surfaces (*A. thaliana* root). In these laboratory evolution setups, 5 to 7 parallel lineages (i.e., subsequent populations with an individual evolutionary trajectory) were followed using 30 to 40 transfers. The detailed descriptions of each experiment were previously published (41–44).

In the Bth_bead study, all evolved lineages displayed significantly enhanced biofilm production accompanied by the appearance of a *B. thuringiensis* fuzzy spreader (FS) colony morphotype variant. This FS variant showed higher competitive ability in most

multicellular traits compared to the ancestral strain, suggesting an important role for diversification during adaptation of *B. thuringiensis* to the abiotic surface biofilm lifestyle. Furthermore, genetic characterization showed that the guanylyltransferase gene was disrupted by an IS element in the FS, which altered the aggregation and hydrophobicity of this variant (41).

In the Bth_root experiments, bacterial lineages displayed enhanced root colonization ability compared with the ancestral strain. Single isolates from two of the evolved lineages showed higher recolonization efficiency of new roots compared with the other lineages, in addition to exhibiting altered bacterial differentiation and pathogenicity. Investigation of a key mutation in the gene encoding the Rho transcription termination factor in these lineages demonstrated how transcriptional rewiring alters cell fate decisions in *B. thuringiensis* (42).

In the case of Bs_pellicle approach, *B. subtilis* diversified into four distinct colony variants that dramatically differed in biofilm formation abilities and expression of biofilm-related genes. Genome comparison suggested that major phenotypic transformations between the morphotypes can be triggered by subtle genetic differences (43).

Finally, in the Bs_root study, *B. subtilis* was shown to rapidly adapt to the *A. thaliana* root environment, several evolved isolates displayed altered colony morphologies. Two selected evolved isolates from independent populations from the final transfer outcompeted the ancestor during root colonization. Re-sequencing of single evolved isolates from independent populations and different time points revealed mutations in genes related to different bacterial traits. The examined evolved isolates also displayed robust biofilm formation in response to plant polysaccharides, impaired motility, and altered growth on plant-derived compounds (44).

In each experiment of these four studies, the evolved clones diversified into distinct pheno- and/or genotypic colony variants in time, including dramatically increased biofilm development of the isolated clones at the end of the experiments (Fig. S1). These findings provided novel insights into how *B. thuringiensis* and *B. subtilis* rapidly adapt to abiotic and biotic surface environment and revealed the evolutionary consequences. However, these four studies only studied the genomic changes of certain selected isolates at selected time points or only focused on certain lineages, while the full understanding of the dynamic evolutionary landscapes was lacking over the full experimental evolution setup. Furthermore, the differences and similarities of adaptive mechanisms in *B. thuringiensis* and *B. subtilis* were not identified when these two species were adapted to the same environment and when one species is adapted to two distinct environments. To overcome these limitations, we revived the archived population samples from these four studies and performed longitudinal whole-population genome sequencing to study the underlying genetic dynamics at high resolution. Generally, parallel evolution was verified in *B. thuringiensis* and *B. subtilis* when adapted to respective selective environments (i.e., mutations were detected in overlapping genes), except in the Bth_root experimental evolution system. Interestingly, we found that transposable elements in *B. thuringiensis* possibly play a critical role in adaptive evolution. Our study provides the first comprehensive mutational landscape of two bacterial species' biofilms that is adapted to an abiotic and biotic surface.

## RESULTS

### Mutation spectrum and genetic diversity

We performed longitudinal whole-population genome sequencing of 69 *B. thuringiensis* and 69 *B. subtilis* samples (subsequent time points of parallel lineages from two evolutionary conditions with each bacterium, see Table S1), as well as the ancestor of each strain, using the DNBSEQ-Tx platform (45, 46). We obtained >180× coverage depth raw data for each sample. The genetic alterations were detected compared with the respective ancestors, including SNPs, Indels (short insertions and deletions), and large fragment insert variants, using the *breseq* pipeline (version 0.35.7) (47, 48). The default parameters called mutations only if they appeared at least two times from each strand

and reached a frequency of at least 5% in the population. Additionally, mutations that were detected in highly polymorphic regions were removed. This analysis revealed 119, 273, 158, and 350 mutations in the evolved lineages from the Bth_bead, Bth_root, Bs_pellicle, and Bs_root conditions, respectively (Table 1; Data Set S1). It is noteworthy that we detected relatively much more mutations in *B. subtilis* when considering the genome size, which is 23.36% smaller than *B. thuringiensis* (4.2 Mb vs 5.5 Mb). Overall and in each lineage of both bacterial species, higher numbers of mutations were detected in the plant root evolved populations compared to the *in vitro* biofilm transfers. Notably, the generation numbers in each experimental evolution approach could not be determined except for Bs_pellicle. This was because we could not determine the exact number of cells colonizing the new root or beads in the other three experimental evolution setups. Nevertheless, large differences have been observed between each condition in the spectrum and the fraction of mutation types (Fig. 1a and b; Table 1; Fig. S2), implying different adaptation strategies of these two species when adapting to these new selective environments. The detected mutations were not distributed uniformly across the populations within each condition, and the large fluctuations in mutation numbers in many populations suggest a process where new mutations displaced older ones (40). Interestingly, a non-synonymous mutation (A50D) was observed in the *mutL* gene reaching a frequency of 46.08% in lineage A of Bth_root at transfer 32 that suggested the potential evolution of a mutator strain. However, relative to other lineages and conditions, no dramatic increase in the number of mutations was observed in this lineage, suggesting that none of experimentally evolving lineages gained a mutator phenotype, at least above the detection limit.

Thirteen cases of transposable element rearrangements were identified in the evolving *B. thuringiensis* lineages (Data Set S1), in contrast to the lack of transposon activity in *B. subtilis*. Importantly, the genome of *B. subtilis* DK1042/NCIB 3610 contains only one incomplete transposase gene in contrast to the 39 transposase genes present in the *B. thuringiensis* 407 Cry- strain.

A large number of intergenic mutations were also detected (Table 1), accounting for 38.63% ± 8.32%, 25.57% ± 4.67%, 24.69% ± 3.46%, and 16.32% ± 5.05% of total mutation in each lineage (Bth_bead, Bth_root, Bs_pellicle, and Bs_root, respectively), many of these likely located in promoter or terminator regions. Again, the relative frequencies of these intergenic region mutations were distinct between abiotic and biotic biofilms. The specific role of these numerous intergenic mutations remains vague in these two species; a unified *B. subtilis* genome and transcriptome annotation atlas are available (49), and recent work has suggested certain intergenic regulatory regions to be critical in *B. subtilis* growth and adaptation in different environment (50–54). Adaptive changes in non-coding regions might possibly affect the binding sites for regulatory proteins. For example,

**TABLE 1** Mutation type and number[d]

| Category | Bth_bead | Bth_root | Bs_pellicle | Bs_root |
|---|---|---|---|---|
| Pseudogene | 5 | 9 | 0 | 7 |
| Non-coding gene[a] | 0 | 3 | 0 | 1 |
| Coding gene | 69 | 184 | 119 | 286 |
| Synonymous | 8 | 27 | 22 | 62 |
| Non-synonymous | 33 | 113 | 69 | 160 |
| Nonsense | 7 | 12 | 4 | 37 |
| InDel[b] | 21 | 32 | 24 | 27 |
| Intergenic[c] | 45 | 77 | 39 | 56 |
| Total number | 119 | 273 | 158 | 350 |
| Fixed | 6 | 47 | 1 | 9 |
| Fixed rate | 5.04% | 17.22% | 0.63% | 2.57% |

[a]Non-coding gene includes tRNA gene, ncRNA gene, and repeat region (IS).
[b]InDel here only refers to the insertion and deletion mutations in coding gene regions.
[c]Intergenic mutations include all kinds of mutations in intergenic region.
[d]There is no overlap between each category.

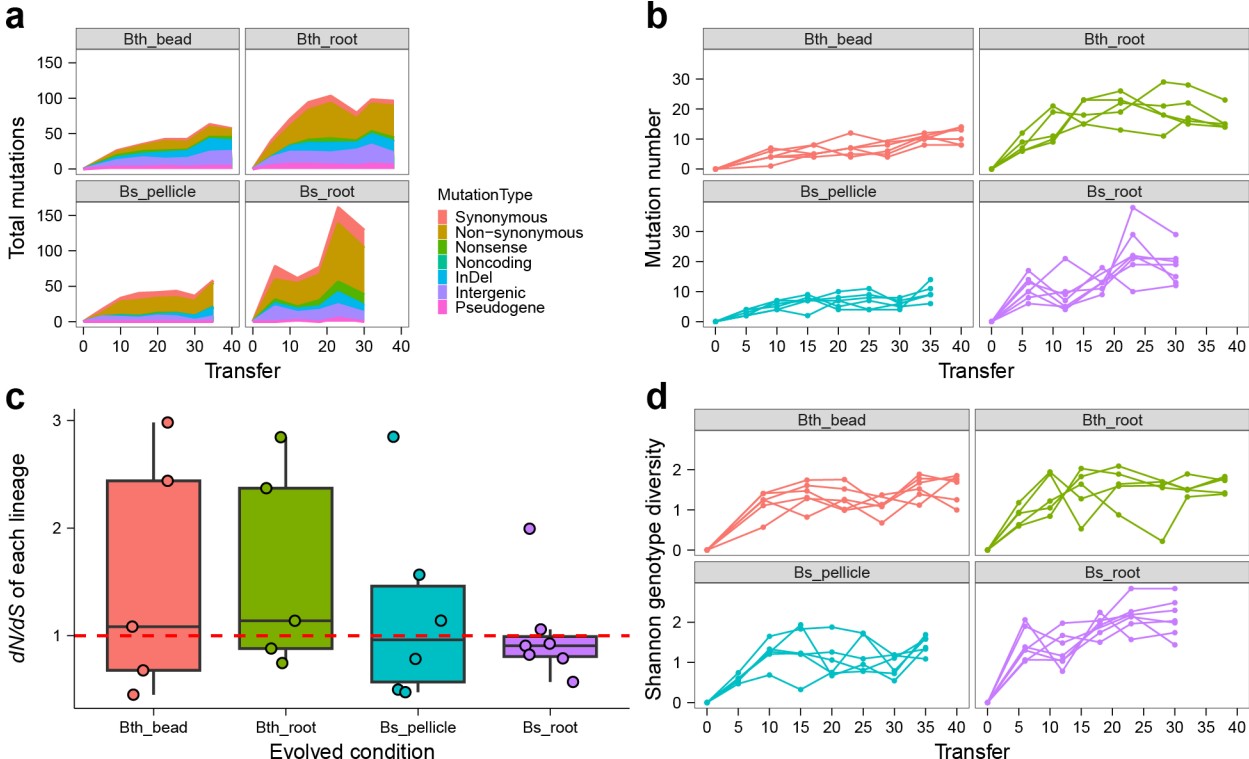

FIG 1 Mutation spectrum, dynamics, and diversity. (a) Distribution of detected mutations of each type in four adaptation models over time. Shaded bars show the distribution of different mutation types in each time point or total. (b) Dynamic distribution of detected mutations in each lineage over time. (c) *dN/dS* ratio. The ratio of non-synonymous to synonymous mutations (*dN/dS*) in the entire pool of detected mutations of each condition, this ratio is normalized by the relative number of synonymous and non-synonymous sites of *B. thuringiensis* and *B. subtilis*, respectively. Boxes indicate Q1–Q3, lines indicate the median, black circles filling with different color indicate the ratio of each lineage. (d) Genotype diversity. Dynamic distribution of genotype alpha diversity in each population of four adaptation models over time calculated using Shannon method. Genotypes and frequencies were generated by *Lolipop* software package in genotype and genealogy analysis.

we found three intergenic point mutations in the region between operon *yhxA-glpP* and operon *glpFK* in four Bs_root lineages, one located at the terminator region of operon *yhxA-glpP* and two at the promoter region of operon *glpFK* (55, 56). These two operons encode proteins essential for growth on glycerol or glycerol 3-phosphate, which is possibly related to the adaptation to glycerol in the medium (57, 58). Five percent of glycerol was supplemented in MSNg medium in Bs_root condition to allow bacteria to initiate growth, before sufficient root exudates or plant polysaccharides are present for the bacterial cells (biofilms) to grow (44, 59). We also found one intergenic SNP in the promoter and another in the terminator region of the gene *sinR*, respectively. The *sinR* gene encodes a DNA-binding transcription factor which is central to the determination of cell fate with respect to biofilm formation versus swimming motility (60). These results implicate adaptive changes in intergenic regions that presumably influence the binding sites for regulatory proteins.

The number of fixed mutations, which reached a frequency of 100% and persisted to the end of the experiment, was higher in *B. thuringiensis* compared with *B. subtilis* (53 vs 10), and the overall mutation frequencies in the *B. thuringiensis* populations were also higher than that in the *B. subtilis* populations (Data Sets S1 and S2), which indicates that either the strength of selection on *B. thuringiensis* was possibly stronger than that on *B. subtilis*, or the population sizes or bottlenecks differed between these two species. Population size estimation revealed a relatively higher population size in *B. subtilis* populations in both biotic and abiotic conditions (Fig. S3). The majority of fixed mutations found in *B. thuringiensis* lineages evolved on the plant roots, and most of them

appeared as cohort, which refers to the appearance and fixation of several mutations in the same lineage at the same time, in contrast to the single fixed mutation identified in *B. subtilis* pellicle transfer approach that otherwise displayed a total of 158 mutations. In addition to the number of fixed mutations and differences in mutation frequencies, the *dN/dS* ratio between each condition was determined. The total *dN/dS* ratio for Bth_bead, Bth_root, Bs_pellicle, and Bs_root conditions was 1.36, 1.25, 0.95 and 0.91, respectively (Fig. 1c; Table S2), suggesting that the majority of the observed mutations in *B. thuringiensis* are adaptive (even those driven extinct by clonal interference) (27). By contrast, the *dN/dS* ratio near or below 1 in *B. subtilis* reflects that selection in favor of beneficial mutations (and presumably typically non-synonymous) is balanced by hitchhiking of neutral mutations and purifying selection against deleterious mutations (30). However, we should notice that some synonymous mutations may have effects on fitness (61), which would reduce the credibility of infer selection pressures from *dN/dS*, although such selection was generally thought extremely weak (62, 63).

We hypothesized that biotic surface adaptation conditions would maintain higher genetic diversity than abiotic conditions due to potentially increased spatial niche heterogeneity and nutritional source difference created by the plant host. Although we detected much more mutations in Bth_root than that in Bth_bead, the genotype alpha diversity (based on the genotype and frequency from genealogy analysis) of the last time point of Bth_root was only slightly higher than that in Bth_bead (genotype alpha diversity, $P = 0.7186$, $t = -0.3731$, $df = 8.1641$ via two-tailed $t$-test) (Fig. 1d; Fig. S4). This non-significant difference was due to the presence of much more mutations in one genotype in Bth_root than in Bth_bead. The Bs_root diversity was higher than that in Bs_pellicle (genotype alpha diversity, $P = 0.008211$, $t = -3.3897$, $df = 8.8412$ via two-tailed $t$-test). We also compared the genotype alpha diversity between the last time point and the first time point sequenced of each condition, revealing a significant difference in all modes, except for in Bth_bead (genotype alpha diversity, $P = 0.0649$, $t = 2.0776$, $df = 9.8394$ via two-tailed $t$-test), this non-significant difference in Bth_bead was highly possibly due to that the first time point sequenced in Bth_bead was from transfer 9, which was already much diversified in genotype. These results again support a process where new mutations were displacing older ones.

## Mobilome activity in experimentally evolving *B. thuringiensis*

As highlighted above, transposition of insertion element in *B. thuringiensis* 407 contributes to its adaptation for colonization of nylon beads and appearance of novel colony morphotypes (41); therefore, we specifically explored potential mobile element rearrangements in the experimentally evolved lineages. We applied a two-round strategy using *breseq* pipeline (v0.35.7) (47, 48) to identify large size transposable element rearrangements (see Materials and Methods). While no IS transposition was exposed in *B. subtilis*, examination of the experimentally evolved lineages revealed 13 IS transposition cases in *B. thuringiensis* (Table 2). In line with higher frequency of IS transposition in experimentally evolved *B. thuringiensis*, closer examination of the genome sequences of the two species exposed 10 types of transposase genes, with 1–12 copies having lengths from 213 to 1,562 bp (four ISs smaller than 900 bp are presumably truncated IS), giving a total of 39 copies in the *B. thuringiensis* 407 genome (Table S3A), in contrast to only one incomplete transposase gene (186 bp) was found in the genome of *B. subtilis* NCIB 3610/DK1042.

Among the 10 types of IS elements observed in *B. thuringiensis* 407, IS*110* family transposase was the most abundant IS with its 12 copies in the ancestor strain, while the IS*4*-like element IS*231*A family transposase (IS*231*A for short below) was the most active element in the experimentally evolved lineages. We identified 12 IS*231*A and 1 IS*110* family transposition cases. Seven genes and three intergenic regions were disrupted by the inserted IS*231*A elements. We found that the cupin domain-containing protein (BTB_RS26870, *rfbM* gene) was disrupted by IS*231*A in three out of the six lineages in Bth_bead with max frequencies from 13.37% to 48.66%, which show high parallelism

**TABLE 2** IS transposition cases detected in the experimentally evolved *B. thuringiensis* lineages

| Position | Insertion sequence[a] | Gene | Description | Condition | Lineage | Highest frequency in lineage |
|---|---|---|---|---|---|---|
| 397,547 | I1(+) + 12 bp | *glnP* | Glutamine transport system permease protein GlnP | Bth_bead | B | 6.12% |
| 607,087 | I1(−) + 11 bp | *npr* | Bacillolysin | Bth_bead | E | 5.23% |
| 985,569 | I1(+) + 11 bp | BTB_RS05005/BTB_RS34715 | --/--- | Bth_bead | A | 34.73% |
| 1,233,591 | I1(−) + 11 bp | *rfbB* | dTDP-glucose 4,6-dehydratase | Bth_bead | A | 54.86% |
| 2,650,349 | I3(+) + 11 bp | BTB_RS13370 | RNaseH domain-containing protein | Bth_bead | D | 8.36% |
| 5,263,506 | I3(+) + 12 bp | *rfbM* | Cupin domain-containing protein | Bth_bead | A | 13.37% |
| 5,263,506 | I3(+) + 12 bp | *rfbM* | Cupin domain-containing protein | Bth_bead | B | 48.66% |
| 5,263,506 | I3(+) + 12 bp | *rfbM* | Cupin domain-containing protein | Bth_bead | F | 26.13% |
| 5,289,559 | I3(−) + 12 bp | *nuoN* | NADH-quinone oxidoreductase subunit N | Bth_bead | A | 29.23% |
| 493,064 | I2(+) + 3 bp | *wbpA* | UDP-N-acetyl-D-glucosamine 6-dehydrogenase | Bth_root | F | 8.59% |
| 524,134 | I1(+) + 11 bp | BTB_RS02650 | GNAT family N-acetyltransferase | Bth_root | C | 100.00% |
| 1,915,321 | I1(−) + 11 bp | *braD* | High-affinity branched-chain amino acid transport system permease protein BraD | Bth_root | F | 18.75% |
| 3,677,033 | I3(+) + 12 bp | *glnR/ynbB* | HTH-type transcriptional regulator GlnR/uncharacterized protein YnbB | Bth_root | E | 12.46% |

[a]The insertion sequence definition can be found in Table S3B.

of adaptation to the plastic bead-attached biofilm environment. We failed to find this mutation in the rest of three lineages in Bth_bead possibly due to low frequency (<5%) of this IS transposition. Importantly, when specifically sequencing FS colony morphotypes, IS*231*A insertion was detected within the *rfbM* gene in each of the six lineages (41). A loss-of-function *rfbM* mutant, Bt407Δ*rfbM*, constructed previously (41), could re-create the FS phenotype and the robust biofilm formation. Additionally, Bth_bead lineage A contained four IS transposition cases with max frequencies from 13.37% to 54.86%.

## Genotype and genealogy analysis of evolving lineages

As mentioned above, hundreds of mutations rose to detectable frequencies (>5%) in these populations. To infer the genealogical structure of each lineages and visualize changes in lineage frequencies from shared, nested mutation trajectories over time, we utilized the *Lolipop* software package developed by Cooper and colleagues (40, 64). We obtained an average of 13.83 ± 2.27, 24 ± 4.29, 17.33 ± 3.25, and 27.43 ± 5.1 genotypes in the Bth_bead, Bth_root, Bs_pellicle, and Bs_root experimental setups, respectively, each with multiple mutations, rose to a detectable frequency. We should notice that the genotype inferred from *Lolipop* might need further correction from isolates data or long reads sequencing data for a more accurate genotype inference and clustering. The number of genotypes in each condition is consistent with the mutation number and diversity within each condition.

### Clonal interference

It is generally expected that a genotype with one or a combination of several beneficial mutations whose combined fitness is superior, may outcompete the other, less fit genotypes. However, due to the numerous possible routes for one bacterium to adapt to a new environment, competition between different genotypes that contain beneficial mutations, the rate at which any one of these genotypes spreads through the population is slowed because it must displace other fitter competitors rather than only its ancestor. Such dynamic population process is called as clonal interference (33, 65–67). In our experiments, we frequently observed clonal interference in all experimentally evolving

lineages (Fig. 2). Numerous mutations or groups of mutations rose to high frequency and then fall to extinction, while outcompeted by another group, and therefore many distinct mutations existed at a very low frequency (below 5%) during the earlier period of experiment and then rose to high frequency in the later period.

We observed nested fixations (68) in lineage F of Bth_bead, lineages A, C, D, and E of Bth_root, and lineage 1 of Bs_root, where beneficial mutations or mutation cohorts fix sequentially in the background of the previous fixed mutation(s) (Fig. 2). Although we observed most nested fixations events in Bth_root, the clonal interference was still very strong in these lineages, especially in the late stage of the experiment.

## Evolvability

During the adaptation experiment of evolving lineages, the selection not only had immediate effects on new mutations or new combinations of alleles but also on those new genotypes that differ in their capability to further evolve (26). The evolvability of the new genotypes may be affected in at least two main ways: changed mutation rates or differences in epistatic interactions with potential further mutations. From the lineage diagram figures (Fig. S5), we observed that there were much more first-order and less second-order genotypes (nested genotypes) in abiotic adaptation conditions than that in biotic adaptation conditions in both bacterial species, while many genotypes persisted along the experiments in the two abiotic adaptation conditions. To better compare the evolvability among different models, we calculate the accumulated mutation numbers of each genotype in each condition at the last time point. The average accumulated mutation number is 2.58, 8.94, 1.75, and 3.1 in Bth_bead, Bth_root, Bs_pellicle, and Bs_root, respectively (Fig. 3). We observed obvious nested fixations in Bth_root, in which the new beneficial mutations accumulated and fixed in the background of the previous ones. Bs_pellicle contained least accumulated mutations. This is possibly caused by the transfer model where the pellicle transfer contained a more homogeneous mix of the

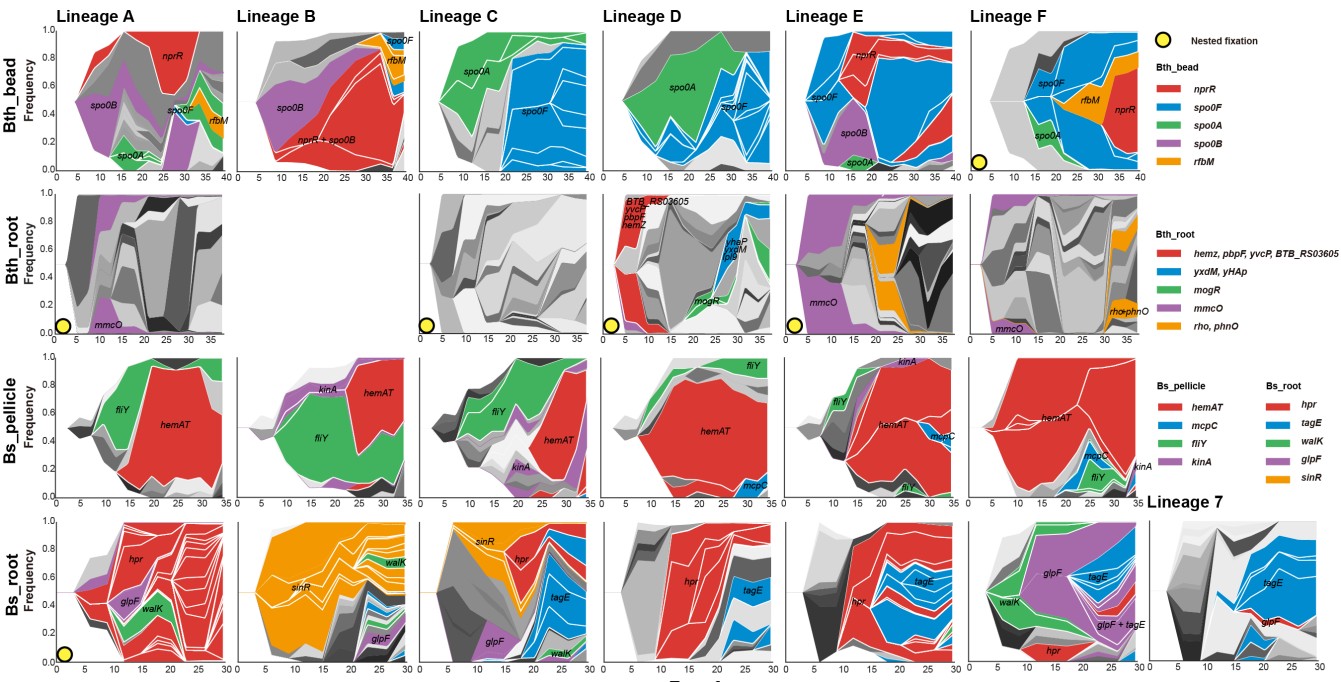

**FIG 2** Genealogy and genotype frequencies over time. Each shade or color represents a different genotype, and vertical area corresponds to genotype frequency, inferred by *Lolipop*. Dominant genotypes that contain the high-frequency mutated genes, which are shared in different populations, are highlighted in certain colors within each adaptation model. The nested genotypes of the dominant genotype are highlighted with the same color in Bth_bead, Bs_pellicle, and Bs_root, except for the nested genotypes which also belong to the dominant genotypes. In Bth_root, for the complex combination of mutations among different lineages and early fix of new mutations, nested genotypes are not highlighted. Other genotypes are in gray.

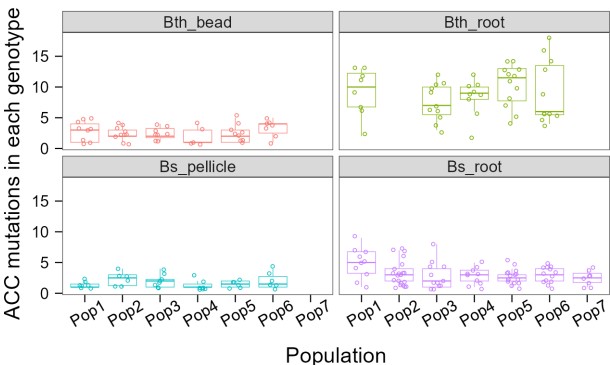

**FIG 3** Evolvability. The number of accumulated mutations in each genotype at the last time point in each condition. Each circle represents one genotype. Boxes indicate Q1–Q3, lines indicate the median, and circles with no filling indicate the accumulated mutation number of each genotype.

biofilm bacterial cells than bead or the root biofilm transfer model. These results showed that due to the evolvability and biofilm surface type, selective conditions have a strong effect on the evolutionary history of bacteria.

### More shared dominance mutated genes or genotypes in abiotic environments

In the Muller plots, which show how diversity emerges and how it changes over time, an obvious pattern could be observed, namely, abiotic environment resulted in a more consistent mutation profile compared with the biotic environment (Fig. 2), which is consistent with the Jaccard index of each environment.

### Parallelism and genetic targets of selection

Next, we quantified parallel evolution by testing whether mutations in certain genes appeared more frequently than expected by chance and calculated the proportion of mutations shared between two evolved lineages within each condition.

We defined "multiplicity" of a gene as the number of mutations in that gene across all sequenced lineages in each condition, scaled by its gene length (27, 30). To exam whether there was parallelism within each species, we also performed this analysis by applying it to both conditions in each species as a whole. Similar to many other laboratory adaptation experiments (30, 33, 63, 64), we observed more high multiplicity genes than expected by chance in each condition and species, relative to a null hypothesis in which mutations are distributed randomly across all genes (Fig. 4a). We also observed that the excess, defined as the degree of observed multiplicity related to the null distribution, was more pronounced in *B. subtilis* than that in *B. thuringiensis*, which is consistent with the number of shared mutated genes in each species. Four genes were detected to be mutated in both *B. thuringiensis* adaptation conditions, while 13 genes were identified in both *B. subtilis* adaptation conditions. Remarkably, in Bth_root, we found that the fraction of genes with multiplicity exceeding the number of lineages in each condition was much lower compared to other conditions (Fig. 4a). Specifically, the number of lineages in Bth_bead, Bth_root, Bs_pellicle, and Bs_root, is 6, 5, 6, and 7, respectively. This was because the five lineages of this experiment could be categorized into three types, each type containing different mutated genes (Fig. 5, left; Data Set S2). When studying the divergence evolution in Bth_root among lineages, we observed three starkly different adaptive strategies that led to marked divergence.

The Jaccard index (*J*) of all pair of lineages (two evolved lineages with sets of accumulated mutated genes G1 and G2) from each evolved condition was calculated. The Jaccard index is a common measure of similarity, which describes the likelihood that the same gene is mutated in two independent lineages (69, 70) and ranges from 0 (no mutated genes in the two lineages are shared, G1∩G2 = 0) to 1 (the two lineages have

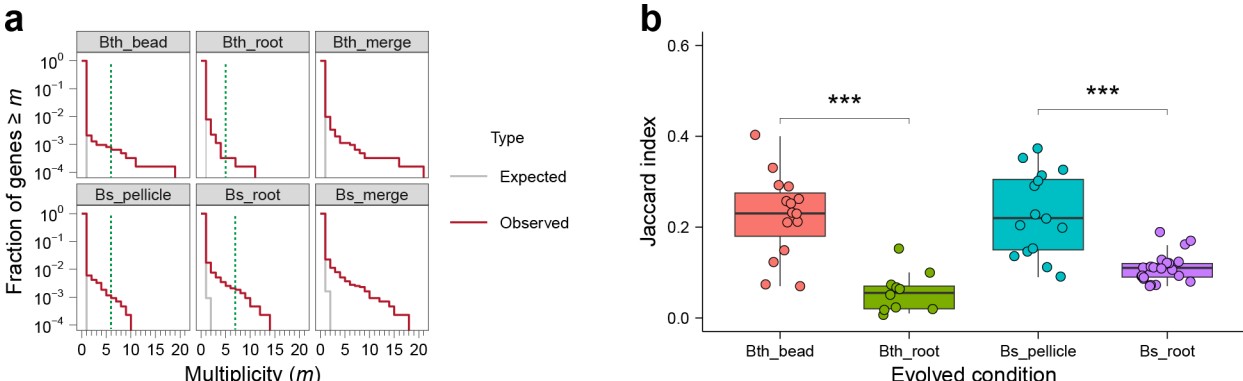

**FIG 4** Parallelism. (a) Parallelism in each adaptation model and species. For all plots, the null distribution (shown in gray) was obtained by simulating random mutations to genes, considering the number of mutations in each model and species in our data and the relative length of each gene. The green dotted line indicates the number of lineages in each condition. (b) Degree of parallelism within each condition estimated by Jaccard index. Asterisks at the top indicate significant differences between biotic and abiotic evolved conditions of each species (***$P < 0.001$, Student's unpaired two-tailed $t$-test was performed). Boxes indicate Q1–Q3, lines indicate the median, black circles filling with different color indicate the $J$ value of each combination, and gray dots indicate the outliers.

exactly the same set of mutated genes, G1 = G2). The $J$ values of abiotic conditions were significantly higher than that of biotic conditions in both species ($P < 0.001$) (Fig. 4b), averaged at 0.22, 0.057, 0.23, and 0.11 for Bth_bead, Bth_root, Bs_pellicle, and Bs_root, respectively, which agrees with the consistency in mutated genes observed previously (Fig. 2). To test whether population size contributes to this parallelism difference (70), we estimate the approximate population size (here we calculate the biofilm production for

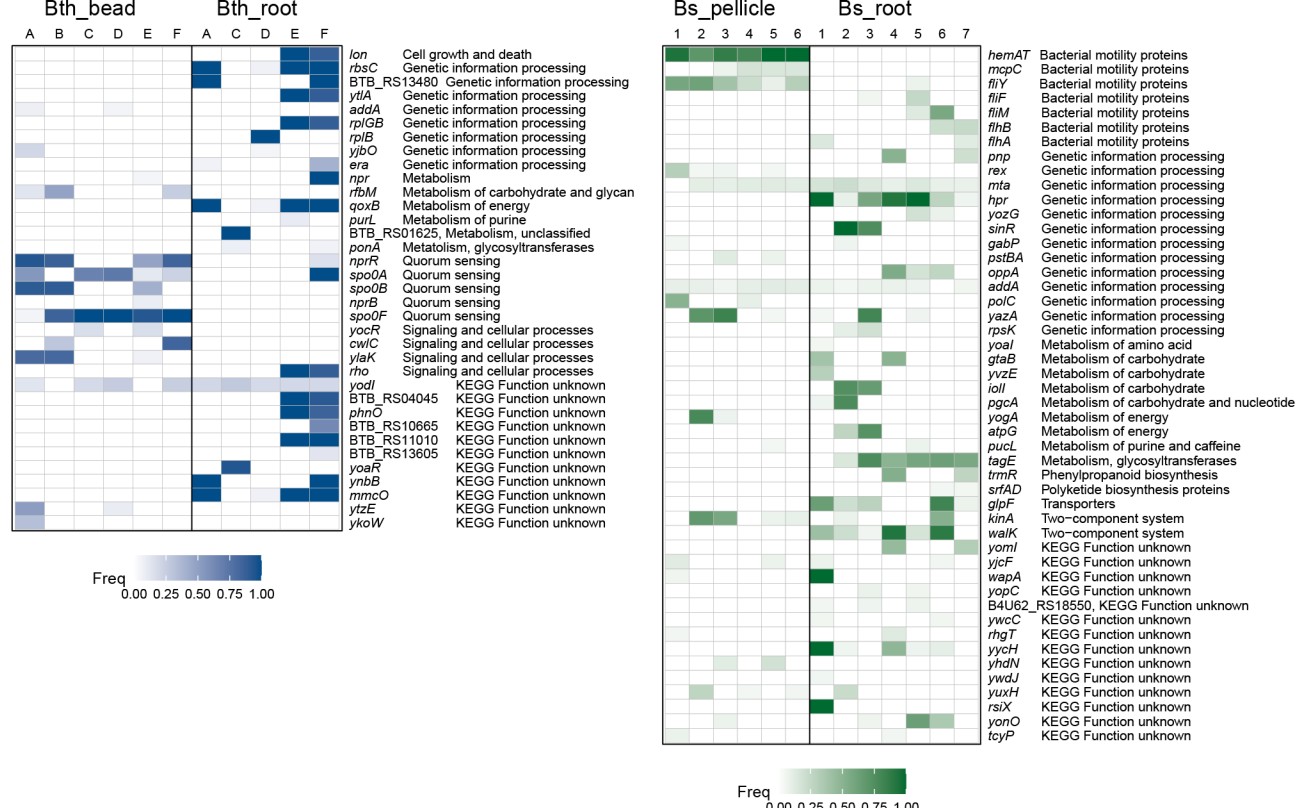

**FIG 5** Genes mutated more than one time in *B. thuringiensis* and *B. subtilis*. Each column represents one replicate lineage. Thirty-five genes and 48 genes mutated more than one time in *B. thuringiensis* (left) and *B. subtilis* (right), respectively, were distributed non-randomly. Color indicates the highest frequency of mutations in that gene in that lineage. Detailed information of these genes can be found in Data Set S2.

comparison) of each condition from the previous data (41–44), ranging from 8.40E+05 to 3.29E+07, 1.13E+03 to 1.05E+06, 3.33E+08 to 1.99E+10, and 6.00E+04 to 4.68E+06 for Bth_bead, Bth_root, Bs_pellicle, and Bs_root, respectively (Fig. S5). The population size of abiotic environments is much larger than that of biotic environments, demonstrating the positive relationship between population size and the probability of parallelism, e.g., that the probability of parallelism increases with population size (70). The small population size of Bth_root also helps to explain why lineage B disappeared only after five transfers during the experimental evolution experiment, and the remaining five lineages of this experiments could be categorized into three evolutionary trajectory types. The strong bottleneck and genetic drift may, therefore, greatly influence the evolutionary landscape of these evolving lineages.

Next, we quantified how signatures of each mutated gene were greater than expected by chance. For this, we focused on genes mutated two or more times in each species. We found that 35 genes in *B. thuringiensis* and 48 genes in *B. subtilis* mutated two or more times, but these genes constituted only 0.88% and 1.89% of the coding genome of *B. thuringiensis* and *B. subtilis*, respectively. Almost all mutations in these genes (except for one gene for each species) were distributed non-randomly (Fisher's exact test and corrected with Benjamini-Hochberg method, $q < 0.05$), accounting for 54 mutations (45.38% of all mutations, Table 1), 57 mutations (20.88%), 81 mutations (51.27%), and 150 mutations (42.86%) in each condition (Bth_bead, Bth_root, Bs_pellicle, and Bs_root, respectively). Most mutations are non-synonymous mutations and indels.

To understand the functional basis of this parallelism, we categorized function of these genes based on KEGG pathway and then focused on the genes that contain non-synonymous and indel mutations which occurred in at least half of the populations in each condition. Mutated genes or genotypes were shared more frequently in abiotic adaptation environments than that in biotic adaptation environments (Fig. 5; Data Set S2).

### Cell fate decision-related genes are most targeted in Bth_bead

In Bth_bead, four quorum sensing and sporulation regulation-related genes (e.g., *spo0F*, *nprR*, *spo0A*, and *spo0B*) were mutated more frequently and reached high frequencies and were even fixed in more than one lineage: *spo0F*(A82T) fixed in lineages C and F, and *spo0F* (Y12*) fixed in lineage D. In addition, *rfbM* gene was disrupted by IS*231*A in three out of the six lineages with max frequencies from 13.37% to 48.66%; variants containing this mutation displayed higher competitive ability compared to the ancestral strain (41).

### Different gene target range observed in Bth_root

In Bth_root, there were a total of 47 fixed mutations across all lineages. Transporter and ribosome-related genetic information processing genes are frequently targeted for selection in Bth_root. Additionally, genes related to motility and metabolism are also selected. As we could observe clearly three different routes of *B. thuringiensis* adapting to the *A. thaliana* root environment, the shared mutated genes could also be separated. Lineage C had no shared mutated genes with other four lineages, except for two shared mutations in very low frequencies with other lineages. Lineage D also had no shared mutated genes with other four lineages, except for four shared mutations in very low frequencies with other lineages. Lineages A, E, and F could be categorized into an independent group of adaptation. Within the lineage A, E, and F groups, lineages E and F shared most of the high parallelism genes, whereas lineage A only shared five genes with lineages E or F. Lineages E and F also contained a *rho* (E54*) mutation, that was studied thoroughly by Lin et al., which demonstrated that the nonsense mutation in *rho* gene impaired the swimming motility of Bt407 but increased robust pellicle formation, and this nonsense mutation in *rho* reshapes the transcriptional landscape of *B. thuringiensis* (42).

## Motility, chemotaxis, and aerotaxis contribute to adaptation in pellicle evolution in Bs_pellicle

In Bs_pellicle, three bacterial motility protein and flagellar motility-related genes, *hemAT*, *fliY*, and *mcpC*, were mutated more frequently than expected and reached to high frequencies or even fixed, supporting a role of these mutations in the adaptation of *B. subtilis* to air-liquid interface environment. We also found mutations in the *kinA* gene, a protein product of the gene that was directly linked to modulating the phosphorylation level of Spo0A, influencing both matrix production and sporulation.

## Multiple adaptation routes of B. subtilis to A. thaliana roots in Bs_root

In Bs_root, we could observe that *tagE*, *hpr (scoC)*, *glpF,* and *walK* were mutated more frequently and reached high frequencies than other genes. Unlike in other conditions, the functions of these genes are not concentrated on a specific biological function but are more dispersed. This suggests the complexity and multiple adaptation routes of *B. subtilis* to *A. thaliana* roots.

## Parallel evolution across species and environments

Besides functional parallel evolution within each model, we also observed a certain parallel evolution across species and environments, especially when two species adapt to the same environment. *B. subtilis* shared more mutated genes in similar KEGG functions than that in *B. thuringiensis* between abiotic and biotic models, which might suggest that the two models in *B. subtilis* are more similar than in *B. thuringiensis*. Additionally, the shared mutated genes in similar KEGG functions are higher for the biotic model of the two microorganisms compared with the abiotic models, suggesting that the functional mechanism for the two species to adapt to the same selection stress is similar, compared to the distinct adaptation to two different abiotic environments (i.e., cyclic colonization of beads by *B. thuringiensis* compared with the artificial disruption of pellicle biofilms to start a new cycle in *B. subtilis*).

## DISCUSSION

We used longitudinal whole-population genome sequencing (27, 28, 30) to study the underlying genetic dynamics of the two species adapt to abiotic and biotic environment at high resolution; this method allowed us to capture the mutations with a frequency of 5%. We observed hundreds of mutations in each evolved condition: there were higher number of mutations, relatively few intergenic mutations, higher fixed mutation rate, and higher genetic diversity in biotic conditions compared with abiotic conditions in both species. In addition, there were more fixed mutations in biotic conditions in both species, and extremely more fixed cases were found in Bth_root which possibly related with the small population size. The *dN/dS* ratio is slightly higher in *B. thuringiensis,* and we only detected IS-mediated mutations in *B. thuringiensis*.

IS elements are small and autonomous transposable elements with variable numbers and copies found in most bacterial genomes and have been reported to play important roles in shaping the genomes of their hosts (23, 71). These simplest transposable elements usually contain only the genes responsible for their transposition (17). The IS-mediated transpositions could lead to gene inactivation or to the activation or alteration of the expression of neighboring genes (22), and consequently, the effect could be either beneficial, deleterious, or neutral. Both the number and activity of the IS elements of a given bacterium will influence the genome structure and gene expression, which will further impact the fitness of the bacterium in certain environments (22, 71). IS-mediated changes have been described to both promote and constrain evolvability of *Escherichia coli* in a long-term evolution experiment (LTEE) (23), and an experimental evolution using cyanobacterium *Acaryochloris marina* revealed that the vast majority of beneficial mutations during laboratory evolution are due to transposition of a single IS element (21). ISs play a critical role in allowing *B. thuringiensis* to adapt to the plastic

bead-attached biofilm environment (41). In fact, we identified a total of 13 IS transposition cases in *B. thuringiensis*. Notably, the *rfbM* gene was disrupted by IS*231*A in three out of six lineages in Bth_bead. The FS variant that was identified previously to harbor this insertion showed altered aggregation and hydrophobicity. Interestingly, this rearrangement of ISs in the genomes of the evolved lineages and final populations' isolates showed high parallelism of adaptation. We also observed four IS-mediated mutations in Bth_root (three mediated by IS*231*A and one mediated IS*110*), while insert positions were distributed in a different area of the genome. This implies that these ISs may only contribute little to adapt to the *A. thaliana*. Our study has important implications for understanding how IS elements in the *B. thuringiensis* genome affect the fitness, biofilm formation, and adaptation.

We frequently observed clonal interference in all experimentally evolving lineages. The frequency of mutations fixation rate was different among the four conditions, and mutations seemed to be more frequent in biotic conditions while also negatively correlated with the population size. When we reviewed the population size of four experiments, we found that the biotic conditions harbored much lower population size than abiotic conditions at the end of each timepoints and each transfer. Our results suggest that a stronger population bottleneck will lead to a more frequent fixation of specific genotypes due to genetic drift. Furthermore, the difference in the frequency of fixed mutations might be explained by the different population sizes, the strong effect of competing beneficial mutations in the abiotic populations, and the strong impact of genetic drift in the biotic populations (72). We also observed nested fixations in some of the evolving lineages, with some even fixed simultaneously as cohort (33, 68), especially in Bth_root (four of five lineages), in which the size of the fixation cohorts was the largest among all conditions (up to seven mutations fixed simultaneously). However, it is very unlikely that these mutations occurred simultaneously when considering the typical mutation rates in bacteria. One possible explanation is that the beneficial mutation(s) occurs in a background that already has certain sets of neutral or deleterious mutation, and therefore, the hitchhiker and beneficial driver mutations fix at the same time (68). We found this possible phenomenon frequently in Bth_root, in lineage D, the *ywdH*: T186T mutation is possibly a neutral mutation that serves as a background. Subsequently, beneficial co-drivers occurred sequentially, leading the genotype containing the cohort to fix. Although *dN/dS* ratios vary among the four conditions, they all approximate a value of 1, which is relatively low compared with the high *dN/dS* ratio observed in previous studies (27, 40). This implies that neutral or weakly deleterious hitchhikers occurred frequently in our experimental evolution models. Thus, the higher fixation rates of mutations (or cohorts) were possibly caused by the smaller population size and strong population bottleneck (genetic drift) under the biotic conditions.

We observed obvious parallelism at genotype level in Bth_bead, Bs_pellicle, and Bs_root conditions, but not in Bth_root, which is consistent with the parallelism at phenotype level. In Bth_root, isolate from lineages E and F displayed improved iterative ecesis on roots and increased virulence against insect larvae. The motility ability, sporulation kinetics, and cell morphologies were also different from other lineages (42). When we reviewed the population size of four experiments, we found that the Bth_root contains the smallest population size. It only contains from 1.13E+03 to 1.05E+06 cells at the end of each timepoints and each transfer. Theory and empirical studies suggest that strong selection and large population sizes increase the probability for parallel evolution at the phenotypic and genotypic levels (33, 70, 73–75). Differences in population sizes between populations exposed to similar conditions can affect the degree of parallelism (73). The use of different-sized experimental populations of the unicellular alga *Chlamydomonas reinhardtii* adapting to a high salt environment demonstrated that adaptation to salt was repeatable at the fitness level in medium and large populations, but not in small populations (74). Similarly, evolution experiments with large and small yeast populations revealed that beneficial mutations occur more consistently in larger populations (33). Furthermore, smaller population size generally

led to a greater among-population variation than large population sizes do in a viral adaptation model (76). As shown above, the abiotic conditions contain much larger population size than that in biotic conditions, which is consistent with the Jaccard index ($J$) and the muller plots; thus, higher degree of parallelisms is observed in the large population size conditions in each species.

Generally, as larger population size permits higher number of mutations, we would expect to detect more mutations, higher diversity, and more intense clonal interference in larger populations for both *B. thuringiensis* and *B. subtilis* in the same environment. In contrast, higher number of mutations and alpha genetic diversity could be detected in total and at the end of the experiment of each lineage in biotic condition in each species in spite of the lower population size under these conditions compared to the abiotic environment. The four experimental evolution setups used here differed in several parameters, e.g., cultivation temperature and volume, from which the biofilm surface, growth temperature, and cultural volume (which would affect nutrition supply) together may play a pivotal role in the adaptation and also possibly affect the population size. Furthermore, we speculate that the population bottleneck size is correlated with the population size, although it was not possible to calculate the exact number of cells transferred from one round to another in these experimental setups. While the *B. subtilis* pellicle population was consecutively diluted 100-fold each time, the generation number could be calculated. We could not determine the exact number of cells colonizing the new root or beads in the other three experimental evolution setups. The strong bottleneck imposed during the transfer process likely influenced the effectiveness of selection, making genetic drift more robust. This was especially evident in Bth_root, which showed the lowest parallelism among the four conditions, both phenotypically and genotypically (72). In addition, a sharper fluctuation of mutation number was also observed in biotic conditions, which suggests that a lot of mutations were lost during transfer process due to the small population bottleneck size in these conditions. The higher number of mutations, higher genotypic diversity, and higher evolvability (accumulated mutations in each genotype at the last timepoint) are puzzling in the populations evolved in the biotic conditions. The averaged mutation number per transfer in each lineage was calculated to be 0.60, 1.20, 0.90, and 1.67 for Bth_bead, Bth_root, Bs_pellicle, and Bs_root, respectively, nearly twice as much in biotic conditions than in abiotic. This is also possibly caused by the stronger bottleneck in biotic conditions, where deleterious and neutral mutations were easier to fix or increase to detectable frequency than that in abiotic conditions with large population size. Furthermore, the spatial niche heterogeneity in biotic conditions may also contribute to the higher genetic diversity within each lineage, while plays a less important role compared to the bottleneck size. In conclusion, our results indicate that both evolutionary (clonal interference) and ecological (population size) factors could potentially influence the genomic landscape of the evolving population of *Bacilli* in distinct environment. Furthermore, the different conditions could affect the degree of adaptation parallelism, while population size is an important driver of evolution.

## MATERIALS AND METHODS

### Experimental evolution

The evolution experiments were performed previously using different models (41–44). **Bth_bead**, *B. thuringiensis* 407 (Bt407 Cry-, commonly referred to as Bt407) ancestor was inoculated into 24-well microtiter plates containing a nylon bead floating in EPS medium with six individual replicates. The plates were then incubated at 30°C with continuous shaking at 90 rpm. After 24 h, the colonized bead was transferred into a fresh EPS medium containing two uncolonized, sterile beads and repeated 40 times (41). **Bth_root** and **Bs_root**, Bt407 and *B. subtilis* DK1042 were inoculated onto *A. thaliana* seedlings into a 48-well plate under hydroponic conditions with six and seven parallel populations, respectively. The 48-well plates were then incubated in a plant chamber (cycles of 16 h

light at 24°C / 8 h dark at 20°C) at mild agitation (90 rpm). Every 48 h, the newly colonized seedling was transferred to a fresh medium containing a new sterile seedling, for a total of 38 and 32 transfers for Bth_root (42) and Bs_root (44), respectively. **Bs_pellicle**, *B. subtilis* DK1042 was grown in MSgg medium statically in a 24-well plate at 30°C for 48 h in six parallel replicates. Then the mature pellicles were harvested, disrupted, and reinoculated into fresh medium after 100× dilution (43). For each experiment, evolved populations at periodic time points were harvested for phenotypic characterization and archived in the freezer to create the resource for further genetic studies.

## Materials

Ancestor bacterial strains and evolved population samples used in this study are from previous studies (41–44). Frozen samples from each adaptation model were revived via cultivation in Lysogeny broth (LB; Lennox, Carl Roth; 10 g·L$^{-1}$ tryptone, 5 g·L$^{-1}$ yeast extract, and 5 g·L$^{-1}$ NaCl) medium for 16 h at 30°C and 37°C for *B. thuringiensis* and *B. subtilis*, respectively, before harvesting the cells for genomic DNA extraction. Genomic DNA was extracted from each culture using EURx Bacterial and Yeast Genomic DNA Kit.

## Sequencing and variant calling

For whole-population sequencing of the evolved populations and ancestor strain samples, acoustic fragmentation PCR-free libraries were prepared using MGIEasy PCR-free Library Prep Set (MGI Tech). Paired-end fragment reads (150 bp × 2) were generated on a DNBSEQ-Tx sequencer (MGI Tech) following the manufacturer's procedures (45, 46). All population samples were sequenced with >180 × depth coverage for polymorphism calling.

Raw data were filtered using SOAPnuke (version 1.5.6) (77) to remove low quality reads: reads including more than 50% of bases with quality lower than 12, reads including more than 10% of unknown base "N," and reads containing adaptor contamination. To ensure the similar variants calling sensitivity, we normalized the clean data to 200× depth except for two *B. thuringiensis* populations with 186.42× and 193.86× depth. Mutations were called using *breseq* (version 0.35.7) with the default parameters and a -p option for population samples (47, 48). The default parameters called mutations only if they appeared at least two times from each strand and reached a frequency of at least 5% in the population. To call large size IS mutations in *B. thuringiensis*, we performed a two-round strategy. First, we ran *breseq* using the default parameters and then annotated the insert sequence (Table S3B) into the reference file manually using the information from unassigned new junction and missing coverage evidence in the first-round *breseq* output (47). Then we performed *breseq* using the new reference again to obtain the IS mutations. The reference genomes used in our analysis are *B. thuringiensis* 407 Cry$^-$ (Bt407) (GenBank accession number CP003889.1) and *B. subtilis* NCIB 3610 genome and pBS plasmid (GenBank accession no. NZ_CP020102 and NZ_CP020103, respectively) for *B. thuringiensis* and *B. subtilis*, respectively. We then removed the mutations that were also found in the ancestor strains and in regions of high polymorphism to obtain the final mutation set for each model.

## *dN/dS* calculation

The neutral ratios of non-synonymous to synonymous substitutions (*dN/dS* ratio) of the ancestral *B. thuringiensis* and *B. subtilis* were estimated by using the codon usage table of each species from Genome2D (http://genome2d.molgenrug.nl/), calculated as 3.69 and 3.51, respectively. All reported *dN/dS* ratios of each adaptation model are normalized by the neutral ratio of synonymous and non-synonymous sites of *B. thuringiensis* and *B. subtilis*, respectively.

$$\frac{dN}{dS} = \frac{\#\text{nonsynonymous mutations}}{\#\text{synonymous mutations}} \div \text{neutral ratio} \qquad (1)$$

## Genotype and genealogy analysis

Muller plots were generated using the *Lolipop* package (https://github.com/cdeitrick/loli-pop) (version 0.6) using default parameters (40, 64). Then Muller plots were manually colored by the presence of shared dominance mutated genes or genotypes.

## Parallelism analysis

We defined "multiplicity" ($m_i$) of gene I as the number of mutations of gene i ($n_i$), multiplied by the mean gene length across all genes ($L_{mean}$), scaled by gene length ($L_i$) (27, 30).

$$m_i = n_i \frac{L_{mean}}{L_i} \qquad (2)$$

In our null model, the mutations for each model or species were randomly assigned to genes from the complete set annotated genes and weighted by gene length of each gene. We simulate these random draws 1,000 times to generate null distributions for gene multiplicity.

To further quantify how signatures of each mutated gene were greater than expected by chance, we focused on genes mutated two or more times in each species. Fisher's exact test was performed using this compiled list: the numbers of mutations per gene, the total number of mutations observed in each species, the gene length, and the genome length of each species. Testing was performed for each gene, and false positives were reduced using the Benjamini-Hochberg correction to adjust the p_value (40).

Jaccard index ($J$) for two evolved lineages with sets of accumulated mutated genes G1 and G2 was calculated using the formula (3). In words, the number of mutated genes shared by both lineages divided by the total number of genes mutated in both lineages.

$$J_{G1,G2} = \frac{|G1 \cap G2|}{|G1 \cup G2|} \qquad (3)$$

## Population size estimation

It is impossible to calculate the exact population size of each population for we only tested the biofilm production in our previous studies, so we calculated the biofilm production as a reference. The experimental evolution setup of Bth_bead contains one old, colonized bead from last transfer and two new colonized beads. The cell number of the new colonized bead was analyzed. Subsequently, we use the cell number multiply by 3 as the population size of Bth_bead from each lineage and timepoint. This method underestimates the population size as it does not count the planktonic cells. We count the biofilm production of each population from transfer 6 to transfer 40.

In both Bth_root and Bs_root, each containsone old, colonized root from last transfer and one new colonized root, and the cell number of the old, colonized root was analyzed. Subsequently, we use the cell number multiply by 2 as the population size of Bth_root and Bs_root from each lineage and timepoint. This method may overestimate the population size since the old, colonized root contains more cells than the new one. We count the biofilm production of each population from transfer 5 to transfer 38 and transfer 4 to transfer 32 for Bth_root and Bs_root, respectively.

In Bs_pellicle, we use the pellicle production directly. This method also underestimates the population size for it did not count the planktonic cells. We count the biofilm production of each population from transfer 5 to transfer 35.

## ACKNOWLEDGMENTS

We thank Anna Dragos, Mathilde Nordgaard, and Yicen Lin for providing the genomic DNA samples from the evolved populations. We thank Jeff Barrick for valuable suggestions on the data analysis method.

This project was supported by China National GeneBank (CNGB), Danish National Research Foundation (DNRF137) for the Center for Microbial Secondary Metabolites, and Novo Nordisk Foundation within the INTERACT project of the Collaborative Crop Resiliency Program (NNF19SA0059360).

## AUTHOR AFFILIATIONS

[1]China National GeneBank, BGI, Shenzhen, China
[2]BGI Research, Shenzhen, China
[3]Bacterial Interactions and Evolution Group, DTU Bioengineering, Technical University of Denmark, Lyngby, Denmark
[4]BGI Research, Beijing, China
[5]Bacterial Ecophysiology and Biotechnology Group, DTU Bioengineering, Technical University of Denmark, Lyngby, Denmark
[6]Shenzhen Key Laboratory of Environmental Microbial Genomics and Application, BGI Research, Shenzhen, China
[7]Institute of Biology, Leiden University, Leiden, The Netherlands

## AUTHOR ORCIDs

Guohai Hu http://orcid.org/0000-0003-2756-8328
Ákos T. Kovács http://orcid.org/0000-0002-4465-1636

## FUNDING

| Funder | Grant(s) | Author(s) |
| --- | --- | --- |
| China National Genebank | | Guohai Hu |
| | | Yue Wang |
| | | Xin Liu |
| | | Bo Wang |
| Danmarks Grundforskningsfond (DNRF) | DNRF137 | Mikael Lenz Strube |
| | | Ákos T. Kovács |
| Novo Nordisk Fonden (NNF) | NNF19SA0059360 | Ákos T. Kovács |

## AUTHOR CONTRIBUTIONS

Guohai Hu, Formal analysis, Investigation, Methodology, Software, Writing – original draft | Yue Wang, Formal analysis, Methodology, Writing – review and editing | Xin Liu, Methodology, Software, Writing – review and editing | Mikael Lenz Strube, Methodology, Writing – review and editing | Bo Wang, Resources, Supervision, Writing – review and editing | Ákos T. Kovács, Conceptualization, Funding acquisition, Methodology, Resources, Supervision, Writing – original draft

## DATA AVAILABILITY

The sequencing data that support the findings of this study have been deposited into CNGB Sequence Archive (CNSA) (79) of China National GeneBank DataBase (CNGBdb) (80) with accession numbers CNP0002416 and CNP0002735.

## ADDITIONAL FILES

The following material is available online.

### Supplemental Material

**Data Set S1 (mSystems00548-23-S0001.xlsx).** Analysis.

**Data Set S2 (mSystems00548-23-S0002.xlsx).** Highest frequency of multi-mutated genes in *B. subtilis*.

**Fig. S1 (mSystems00548-23-S0003.pdf).** Biofilm productivity of four experiments.

**Fig. S2 (mSystems00548-23-S0004.pdf).** Mutation spectrum.

**Fig. S3 (mSystems00548-23-S0005.pdf).** Estimated population size of each population and timepoints calculated from biofilm productivity data.

**Fig. S4 (mSystems00548-23-S0006.pdf).** Genotype diversity.

**Fig. S5 (mSystems00548-23-S0007.pdf).** Linage diagrams.

**Table S1 (mSystems00548-23-S0008.pdf).** Laboratory evolution setups of four experiments.

**Table S2 (mSystems00548-23-S0009.pdf).** dN/dS ratio.

**Table S3 (mSystems00548-23-S0010.pdf).** Transposase gene in the *B. thuringiensis* 407 genome and Insertion sequence definition in the second-round analysis using breseq.

## Open Peer Review

**PEER REVIEW HISTORY (review-history.pdf).** An accounting of the reviewer comments and feedback.

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
