## [Reviewer comments · mSystems]

Species and condition shape the mutational spectrum in experimentally evolved biofilms

Guohai Hu, Yue Wang, Xin Liu, Mikael Strube, Bo Wang, and Ákos T Kovács

Corresponding Author(s): Ákos T Kovács, Universiteit Leiden

Review Timeline:

Submission Date:	May 27, 2023
Editorial Decision:	June 26, 2023
Revision Received:	August 8, 2023
Accepted:	August 11, 2023

Editor: Benjamin Wolfe

Reviewer(s): The reviewers have opted to remain anonymous.

Transaction Report:

DOI: <https://doi.org/10.1128/msystems.00548-23>

June 26, 2023

Prof. Ákos T Kovács
Universiteit Leiden
Institute of Biology Leiden
Sylviusweg
Leiden 2333BE
Netherlands

Re: mSystems00548-23 (Species and condition shape the mutational spectrum in experimentally evolved biofilms)

Dear Prof. Ákos T Kovács:

Thank you for submitting your manuscript to mSystems. We have completed our review and I am pleased to inform you that, in principle, we expect to accept it for publication in mSystems. However, acceptance will not be final until you have adequately addressed the reviewer comments.

Preparing Revision Guidelines

Please return the manuscript within 60 days; if you cannot complete the modification within this time period, please contact me. If you do not wish to modify the manuscript and prefer to submit it to another journal, please notify me of your decision immediately so that the manuscript may be formally withdrawn from consideration by mSystems.

Sincerely,

Benjamin Wolfe

Senior Editor, mSystems

Reviewer comments:

Reviewer #1 (Comments for the Author):

The authors did an adequate job addressing the revisions. That said, the authors should be aware that their revisions on lines 519-520 "The small (strong) bottleneck imposed..." have made the sentence even more confusing. The authors should just say, "The strong bottleneck".

Reviewer #2 (Comments for the Author):

In this study, the authors apply longitudinal whole-population sequencing to an evolution experiment following biofilm-forming populations of two *Bacillus* species [*B. subtilis* (B.s) and *B. thuringiensis* (B.t.)] in two conditions [Arabidopsis roots, and lab culture (pellicle/bead)]. The authors find that the populations from the root environment contain a greater overall number of mutations and a greater number of fixed mutations. The authors also found that *B. s.* had a faster evolutionary rate and accumulated more mutations than *B.t.*, especially considering the smaller genome of *B.s.* Moreover, because *B.t.* contains active IS elements, IS transposition played a greater role in *B.t.* evolution. Of the mutations arising in the evolving populations, genes annotated as "Genetic information processing" were common mutation targets for plant root populations. Alternatively, *B.t.* bead populations often evolved mutations in cell-fate behavior genes; and *B.s.* pellicle populations often evolved mutations in motility genes.

The authors conclude that there is a more general adaptive strategy in the biotic conditions between species, but adaptation is more species-specific in the abiotic conditions. This should be somewhat expected as the abiotic conditions are much different between species.

The mutational profiling looks much better in the revised version, but I have some concerns about the calculation and interpretation of dN/dS. I also see some issues with relying on the Lolipop program for inferences of population structure and genotypic lineage (detailed in the comments below). I encourage the authors to re-evaluate some of the inferences made with Lolipop. For the populations that you have clonal sequencing (I believe the authors state in the introduction that they previously published clonal sequencing from these populations), you may be able to perform clone-informed resolution of population structure. Here, one could check instances where it's known that two mutations are linked within a clone and confirm that Lolipop is also inferring those mutations to be from the genotypic lineage. Overall, I don't believe the relative relationships between the two species/conditions will change, but I do believe the size of some of the diversity estimates may decrease and it could change the interpretation of the evolvability result section.

My specific comments are as follows:

1) Line 168-172: "It is noteworthy that we detected relatively much more mutations in *B. subtilis* when we considering the genome size, which is 23.36% smaller than *B. thuringiensis* (4.2Mb vs 5.5Mb). Overall and in each lineage of both bacterial species, higher numbers of mutations were detected in the plant root evolved populations compared to the in vitro biofilm transfers" I understand from the authors' response to Reviewer 1 that they are unable to estimate the differences in the number of generations or true population bottleneck size between the experimental treatments, which is why the interpretation of these results is left a bit open starting on line 211: "Which indicates that either the strength of selection on *B. thuringiensis* was possibly stronger than that on *B. subtilis*, or the population sizes or bottlenecks differed between these two species". However, estimates of population size are reported far later in the results on lines 353-357 and used as an example of selection being able to more work more efficiently and influence parallelism. It might be helpful to present population size earlier to add context to the mutation/evolutionary rate results.

Similarly, there is not much information on how the environmental conditions differ until the end of the discussion. Even then it's vague, with mention of temperature and volume differences. But since the environment can directly impact mutation rate and spectrum, how the environments differ is helpful context for interpreting these results. Especially if these environments are stressful in any way (pH, oxidative, etc.), or if the nutritional differences result in one species spending more time in stationary phase, this can all increase the mutation rate and alter the mutation spectrum. Tom Ferenci's group has a nice paper on this in *E. coli* (<https://journals.plos.org/plosbiology/article?id=10.1371/journal.pbio.2001477>). Is there any sense of how the different factors associated with these environments could affect base-genetic mutation rate and thus the mutation spectra that the authors observed?

2) Line 172-173: "Notably, the generation numbers in each experimental evolution approach could not be determined." Maybe just a one-sentence explanation of why generation numbers could not be determined. This would help the reader appreciate the context of this comment.

3) Line 190- 193: "A large number of intergenic mutations were also detectedmany of these likely located in promoter or terminator regions." This can be somewhat determined for *B. subtilis* at least. <https://rth.dk/resources/bsgatlas/geneset.php?group=Operons> I don't know if there is a similar resource for *B. thuringiensis*, or if their operon structure is similar to *B. subtilis*.

4) Lines 201-203: "These two operons encode proteins essential for growth on glycerol or glycerol 3-phosphate, which is possibly related to the adaptation to glycerol in the medium." From reading the methods section in this paper, it's unclear why glycerol would be in the root adaptation medium. After going through the references, I was able to find that "The MSNg medium used in the EE is a minimal medium supplemented with a very low concentration of glycerol (0.05%)". As such, the authors could still add even more clarification to the methods of how the experimental evolution was conducted. The authors do include additional context on how the media/environments differ in the discussion, but some more context up front would help to understand the adaptive patterns that were observed in this study, especially when these differences are going to be highlighted as a result.

5) Lines 203-204: Are the mutations in *sinR* only found in the *Bacillus subtilis* root treatment? As the section is currently written, it's a bit unclear.

6) Lines 214 - 216: Inferring selection coefficients from dN/dS over short-time scales in clonal populations is problematic. First, short-time scales and clonal reproduction already violate the Wright-Fisher model upon which dN/dS is based. But strong hitchhiking effects from sweeps in the absence of recombination can break the relationship between dN/dS and selection coefficients. As the majority of the fixed mutations in the Bt Root treatment swept together, I would interpret dN/dS ratios with caution. (<https://doi.org/10.7554/eLife.48714> ; <https://doi.org/10.1371/journal.pgen.1000304>). Moreover, there is increasing evidence that synonymous mutations in microbial experimental evolution populations are not as neutral as once believed. Here, codon preference can have measurable fitness effects and again skew the relationship between dN/dS and selection coefficients (<https://www.nature.com/articles/ncomms5076>).

7) Sections: Alpha Diversity measurement, Genotype and genealogy analysis of evolving lineages. Clonal interference. Evolvability. I am struggling with many of the results in this section as they rely completely on the Lolipop analysis which has some concerning shortcomings when trying to resolve rare and transient alleles. These shortcomings can be best visualized in the lineage plots from supplemental figure 4. Let's take Figure S4A, lineage 1 for example. If we believe that lineage 1 is comprised of primarily independent lineages and that Lolipop has resolved genealogy correctly, then this results in a summed genotype frequency of >1 at the final timepoint which is not possible (using the allele frequencies from Dataset S1):
Nested lineage of genotype 7, 9, 11 with a parent mutation of Spo0A I58N (AF at T6 = 0.55);
Lineage of genotype 3 (pseudoSNP, AF at T6 = 0.45)
Nested lineage parented by genotype 16,15,14,10 (IntSNP:985575; AF at T6 = 0.28);
Lineage of genotype 6 (ylaK; AF at T6 = 0.12)
Total AF = 1.4

As such, I recommend caution in using these inferences of population structure as the number of distinct genotypes is likely to be overestimated, and many genotypes may be inferred by lolipop to be independent when they are actually nested. This would inflate the number of what the authors refer to as first-order genotypes, and underpredict the number of second-order genotypes.

Reviewer #1 (Comments for the Author):

The authors did an adequate job addressing the revisions. That said, the authors should be aware that their revisions on lines 519-520 "The small (strong) bottleneck imposed..." have made the sentence even more confusing. The authors should just say, "The strong bottleneck".

Our response: Thanks for the suggestions, Adjustments have been made in the revised manuscript.

Reviewer #2 (Comments for the Author):

In this study, the authors apply longitudinal whole-population sequencing to an evolution experiment following biofilm-forming populations of two *Bacillus* species [*B. subtilis* (B.s) and *B. thuringiensis* (B.t.)] in two conditions [Arabidopsis roots, and lab culture (pellicle/bead)]. The authors find that the populations from the root environment contain a greater overall number of mutations and a greater number of fixed mutations. The authors also found that B. s. had a faster evolutionary rate and accumulated more mutations than B.t., especially considering the smaller genome of B.s. Moreover, because B.t. contains active IS elements, IS transposition played a greater role in B.t. evolution. Of the mutations arising in the evolving populations, genes annotated as "Genetic information processing" were common mutation targets for plant root populations. Alternatively, B.t. bead populations often evolved mutations in cell-fate behavior genes; and B.s. pellicle populations often evolved mutations in motility genes.

The authors conclude that there is a more general adaptive strategy in the biotic conditions between species, but adaptation is more species-specific in the abiotic conditions. This should be somewhat expected as the abiotic conditions are much different between species.

The mutational profiling looks much better in the revised version, but I have some concerns about the calculation and interpretation of dN/dS. I also see some issues with relying on the Lolipop program for inferences of population structure and genotypic lineage (detailed in the comments below). I encourage the authors to re-evaluate some of the inferences made with Lolipop. For the populations that you have clonal sequencing (I believe the authors state in the introduction that they previously published clonal sequencing from these populations), you may be able to perform clone-informed resolution of population structure. Here, one could check instances where it's known that two mutations are linked within a clone and confirm that Lolipop is also inferring those mutations to be from the genotypic lineage. Overall, I don't believe the relative relationships between the two species/conditions will change, but I do believe the size of some of the diversity estimates may decrease and it could change the interpretation of the evolvability result section.

Our response: Many thanks for the positive feedback on our manuscript. The critics have been very valuable and appreciated and ultimately helped to improve the manuscript. We have adjusted the manuscript according to the reviewer's suggestions (more specifically answered in the response to comment 2 just below). Unfortunately, the previous genome re-sequencing efforts of clones all concentrated on limited number of isolates from each population / some populations and these have been isolated at the end point of each evolution experiment.

My specific comments are as follows:

1) Line 168-172: "It is noteworthy that we detected relatively much more mutations in *B. subtilis* when we considering the genome size, which is 23.36% smaller than *B. thuringiensis* (4.2Mb vs 5.5Mb). Overall and in each lineage of both bacterial species, higher numbers of mutations were detected in the plant root evolved populations compared to the in vitro biofilm transfers" I understand from the authors' response to Reviewer 1 that they are unable to estimate the differences in the number of generations or true population bottleneck size between the experimental treatments, which is why the interpretation of these results is left a bit open starting on line 211: "Which indicates that either the strength of selection on *B. thuringiensis* was possibly stronger than that on *B. subtilis*, or the population sizes or bottlenecks differed between these two species". However, estimates of population size are reported far later in the results on lines 353-357 and used as an example of selection being able to more work more efficiently and influence parallelism. It might be helpful to present population size earlier to add context to the mutation/evolutionary rate results.

Similarly, there is not much information on how the environmental conditions differ until the end of the discussion. Even then it's vague, with mention of temperature and volume differences. But since the environment can directly impact mutation rate and spectrum, how the environments differ is helpful context for interpreting these results. Especially if these environments are stressful in any way (pH, oxidative, etc.), or if the nutritional differences result in one species spending more time in stationary phase, this can all increase the mutation rate and alter the mutation spectrum. Tom Ferenci's group has a nice paper on this in *E. coli* (<https://journals.plos.org/plosbiology/article?id=10.1371/journal.pbio.2001477>). Is there any sense of how the different factors associated with these environments could affect base-genetic mutation rate and thus the mutation spectra that the authors observed?

Our response: Many thanks for the suggestion and raising the later question. As for the first suggestion, we have added the population size earlier to the context. For the second question, we aim to discuss it more specifically.

Environmental stresses can increase genetic variation in bacteria, different nutritional stresses can affect the mutation rate, mutation spectrum as well as selection (1). Within a normal (non-stressful) range, increased temperature will increase spontaneous mutation rate (2, 3), higher temperatures can lead to faster evolutionary processes due

to shorter generation times, increased mutation rates and accelerated natural selection (3). Additionally, as reviewed by Michael Lynch et al., there is a significant negative relationship between mutation rate (u) and effective population size (N_e) in bacteria, which means larger population size with lower mutation rate (4).

In our study, three different mediums were used with different cultural temperatures as shown in Table S1. Briefly, **MSNg** medium was used in two biotic evolved conditions, MSNg allows bacteria to initiate growth before sufficient root exudates or plant polysaccharides are present for the bacterial cells (biofilms) to grow(5). **MSgg** medium, a minimal medium that promotes biofilm formation in *B. subtilis* was used in Bs_pellicle and **EPS** medium, which was low-nutrient medium specifically for *B. cereus* biofilm formation was used in Bth_bead. And the temperature used in two abiotic evolved conditions were relatively higher than that in two biotic evolved conditions (30 °C versus 16 h light at 24 °C/8 h dark at 20 °C), it is reasonable to expect a higher mutation rate in two abiotic conditions if the same adaptation model was used, while it was not the case as there were less mutations revealed in these conditions, which indicating that temperature alone cannot explain the differences in number of revealed mutations and the importance of the adaptation model used in our experiments. We speculate that the higher temperature would lead to relatively shorter generation time and then larger population size in abiotic conditions. However, the spatial niche heterogeneity caused by the root exudates and biotic surface, and the strong bottleneck in biotic conditions may together lead to deleterious and neutral mutations were easier to fix or increase to detectable frequency than that in abiotic conditions with large population size.

Adjustment has been made in the revised manuscript (Line 219-221).

2) Line 172-173: "Notably, the generation numbers in each experimental evolution approach could not be determined." Maybe just a one-sentence explanation of why generation numbers could not be determined. This would help the reader appreciate the context of this comment.

Our response: Many thanks for the suggestion, adjustment has been made in the revised manuscript (Line 174-176).

3) Line 190- 193: "A large number of intergenic mutations were also detectedmany of these likely located in promoter or terminator regions." This can be somewhat determined for *B. subtilis* at least. <https://rth.dk/resources/bsgatlas/geneset.php?group=Operons> I don't know if there is a similar resource for *B. thuringiensis*, or if their operon structure is similar to *B. subtilis*.

Our response: Many thanks for the suggestion and the useful information. We didn't find a similar resource for *B. thuringiensis*. Adjustment has been made in the revised manuscript (Line 197-198).

4) Lines 201-203: "These two operons encode proteins essential for growth on glycerol or glycerol 3-phosphate, which is possibly related to the adaptation to glycerol in the medium." From reading the methods section in this paper, it's unclear why glycerol would be in the root adaptation medium. After going through the references, I was able to find that "The MSNg medium used in the EE is a minimal medium supplemented with a very low concentration of glycerol (0.05%)". As such, the authors could still add even more clarification to the methods of how the experimental evolution was conducted. The authors do include additional context on how the media/environments differ in the discussion, but some more context up front would help to understand the adaptive patterns that were observed in this study, especially when these differences are going to be highlighted as a result.

Our response: Many thanks for your valuable suggestions. 5% of glycerol was supplemented in MSNg medium, which allows bacteria to initiate growth, before sufficient root exudates or plant polysaccharides are present for the bacterial cells (biofilms) to grow (5, 6).

Adjustment has been made in the revised manuscript (Line 206-208).

5) Lines 203-204: Are the mutations in *sinR* only found in the *Bacillus subtilis* root treatment? As the section is currently written, it's a bit unclear.

Our response: Many thanks for raising this question. In the current study, the mutations in *sinR* were only found in the *B. subtilis* root treatment. The *sinR* gene encodes biofilm master regulator (7), which regulate the expression of the operons involved in biofilm matrix production, *epsA-O* and *tapA-sipW-tasA* (8, 9). Besides found in root treatment conditions(6, 10), mutations in *sinR* were also found in LB plate evolved conditions(11, 12). Wrinkled colony-forming evolved isolates were found to possess point mutations in *sinR*, and the evolved biofilm phenotypes can be restored by introducing these mutations into the ancestral strain, which suggesting that these mutations are sufficient for robust biofilm formation in wrinkled colony types(12). We speculate that the changes in *sinR* might be an adaptive strategy of *B. subtilis* which could improve its biofilm formation ability.

6) Lines 214 - 216: Inferring selection coefficients from dN/dS over short-time scales in clonal populations is problematic. First, short-time scales and clonal reproduction already violate the Wright-Fisher model upon which dN/dS is based. But strong hitchhiking effects from sweeps in the absence of recombination can break the relationship between dN/dS and selection coefficients. As the majority of the fixed mutations in the Bt Root treatment swept together, I would interpret dN/dS ratios with caution.

(<https://doi.org/10.7554/eLife.48714>);
(<https://doi.org/10.1371/journal.pgen.1000304>). Moreover, there is increasing evidence

that synonymous mutations in microbial experimental evolution populations are not as neutral as once believed. Here, codon preference can have measurable fitness effects and again skew the relationship between dN/dS and selection coefficients (<https://www.nature.com/articles/ncomms5076>).

Our response: Thank you for expressing these concerns. We are uncertain of the first question was correctly understood by us, we made our best to answer this concern.

We transferred the population samples 30-40 times in each experiment, which is equivalent to hundreds of generations, and it may take animals and plants thousands of years to pass similar generations, which is a very long-times scale. To infer selection pressure in protein-coding genes by calculating the evolutionary rate ratio dN/dS is the oldest and most widely used method(13), it has been widely used to study the evolution and selection of bacteria populations in diverse conditions(14–17), e.g, the *E. coli* LTEE(17). However, as the reviewer suggested, the relationship between selection and dN/dS does not follow a monotonic function under some conditions(13, 18), e.g., when effect of synonymous mutations on fitness were unneutral and hence purifying selection acts on synonymous changes(13). While it is generally thought that such selection is extremely weak, affects only a small fraction of sites at risk for synonymous mutations, or both(19–21). In our study, as mentioned in the manuscript, “Although there are differences in dN/dS among four conditions, the total ratio are all near 1, which is relatively low compared with the high dN/dS ratio observed in previous studies(16, 17)”, which further emphasizes the role of population size and genetic drift in our study, but not selection, especially in two biotic evolved conditions.

We agree with the reviewer that we should interpret dN/dS ratios with caution. We now refer in the manuscript of the above-mentioned publication highlighting the potential influence of synonymous mutations (Line 232-235).

7) Sections: Alpha Diversity measurement, Genotype and genealogy analysis of evolving lineages. Clonal interference. Evolvability. I am struggling with many of the results in this section as they rely completely on the *Lolipop* analysis which has some concerning shortcomings when trying to resolve rare and transient alleles. These shortcomings can be best visualized in the lineage plots from supplemental figure 4. Let's take Figure S4A, lineage 1 for example. If we believe that lineage 1 is comprised of primarily independent lineages and that *Lolipop* has resolved genealogy correctly, then this results in a summed genotype frequency of >1 at the final timepoint which is not possible (using the allele frequencies from Dataset S1):

Nested lineage of genotype 7, 9, 11 with a parent mutation of Spo0A I58N (AF at T6 = 0.55);

Lineage of genotype 3 (pseudoSNP, AF at T6 = 0.45)

Nested lineage parented by genotype 16,15,14,10 (IntSNP:985575; AF at T6 = 0.28);

Lineage of genotype 6 (ylaK; AF at T6 = 0.12)

Total AF = 1.4

As such, I recommend caution in using these inferences of population structure as the number of distinct genotypes is likely to be overestimated, and many genotypes may be inferred by *lollipop* to be independent when they are actually nested. This would inflate the number of what the authors refer to as first-order genotypes, and underpredict the number of second-order genotypes.

Our response: Many thanks for raising these concerns. As you mentioned, genotype diversity, genotype inference and evolvability analysis were based on results from *Lollipop* using the default parameter (16, 22, 23). The linkage relationship of different mutations in the population samples are extremely complicated, some hotspot mutations might occur at different genetic backgrounds in one population, which might cause the high total frequency at last time point. Since we only have the short read sequencing data, it is unlikely to infer all the exact linkage between mutations. To infer the genealogical structure of each lineage using *Lollipop* have provided us a relative optimal way to visualize changes in genotype frequencies over time, although with some shortcomings as the reviewer mentioned. We have checked whether genotypes / mutations were coupled properly by checking the available sequencing data from previous chosen isolates. We found that only very limited genotypes / mutations were probably coupled incorrectly, indicating that these results wouldn't affect the trends and conclusion. It would be better to check the validity of *Lollipop* more systematically, which would need more clone samples from different time points of each lineage. It is necessary to develop better bioinformatic tools for more accurate genotype and its frequency inference or could perform long reads sequencing for population samples in Experimental Evolution in the future.

We have mentioned the potential shortcomings of the method in the revised manuscript (Line 288-290).

Reference

1. Maharjan RP, Ferenci T. 2017. A shifting mutational landscape in 6 nutritional states: Stress-induced mutagenesis as a series of distinct stress input–mutation output relationships. *PLoS Biol* 15:e2001477.
2. LINDGREN D. 2009. The temperature influence on the spontaneous mutation rate. *Hereditas* 70:165–177.
3. Chu XL, Zhang BW, Zhang QG, Zhu BR, Lin K, Zhang DY. 2018. Temperature responses of mutation rate and mutational spectrum in an *Escherichia coli* strain and the correlation with metabolic rate. *BMC Evol Biol* 18:1–8.
4. Lynch M, Ackerman MS, Gout JF, Long H, Sung W, Thomas WK, Foster PL. 2016. Genetic drift, selection and the evolution of the mutation rate. *Nat Rev Genet* 17:704–714.
5. Beauregard PB, Chai Y, Vlamakis H, Losick R, Kolter R. 2013. *Bacillus subtilis* biofilm induction by plant polysaccharides. *Proc Natl Acad Sci U S A* 110:E1621–E1630.
6. Nordgaard M, Blake C, Maróti G, Hu G, Wang Y, Strube ML, Kovács ÁT. 2022. Experimental evolution of *Bacillus subtilis* on *Arabidopsis thaliana* roots reveals fast adaptation and improved root colonization. *iScience* 25:104406.
7. Kearns DB, Chu F, Branda SS, Kolter R, Losick R. 2005. A master regulator for biofilm formation by *Bacillus subtilis*. *Mol Microbiol* 55:739–749.
8. Kearns DB. 2010. A field guide to bacterial swarming motility. *Nat Rev Microbiol*. Nature Publishing Group <https://doi.org/10.1038/nrmicro2405>.
9. Chu F, Kearns DB, Branda SS, Kolter R, Losick R. 2006. Targets of the master regulator of biofilm formation in *Bacillus subtilis*. *Mol Microbiol* 59:1216–1228.
10. Hu G, Wang Y, Blake C, Nordgaard M, Liu X, Wang B, Kovács ÁT. 2023. Parallel genetic adaptation of *Bacillus subtilis* to different plant species. *Microb Genom* 9:2023.03.17.533125.
11. Richter A, Blei F, Hu G, Schwitalla JW, Lozano-Andrade CN, Jarmusch SA, Wibowo M, Kjeldgaard B, Surabhi S, Jautzus T, Phippen CBW, Tyc O, Arentshorst M, Wang Y, Garbeva P, Larsen TO, Ram AFJ, Hondel CAM van den, Maróti G, Kovács ÁT. 2023. Enhanced niche colonisation and competition during bacterial adaptation to a fungus. *bioRxiv* 2023.03.27.534400.
12. Leiman SA, Arboleda LC, Spina JS, McLoon AL. 2014. SinR is a mutational target for fine-tuning biofilm formation in laboratory-evolved strains of *Bacillus subtilis*. *BMC Microbiol* 14:301.

13. Spielman SJ, Wilke CO. 2015. The Relationship between dN/dS and Scaled Selection Coefficients. *Mol Biol Evol* 32:1097–1108.
14. Johnson MS, Gopalakrishnan S, Goyal J, Dillingham ME, Bakerlee CW, Humphrey PT, Jagdish T, Jerison ER, Kosheleva K, Lawrence KR, Min J, Moulana A, Phillips AM, Piper JC, Purkanti R, Rego-Costa A, McDonald MJ, Ba ANN, Desai MM. 2021. Phenotypic and molecular evolution across 10,000 generations in laboratory budding yeast populations. *Elife* 10:e63910.
15. Lieberman TD, Michel JB, Aingaran M, Potter-Bynoe G, Roux D, Davis MR, Skurnik D, Leiby N, Lipuma JJ, Goldberg JB, McAdam AJ, Priebe GP, Kishony R. 2011. Parallel bacterial evolution within multiple patients identifies candidate pathogenicity genes. *Nat Genet* 43:1275–1280.
16. Harris KB, Flynn KM, Cooper VS. 2021. Polygenic Adaptation and Clonal Interference Enable Sustained Diversity in Experimental *Pseudomonas aeruginosa* Populations. *Mol Biol Evol* 38:5359–5375.
17. Good BH, McDonald MJ, Barrick JE, Lenski RE, Desai MM. 2017. The dynamics of molecular evolution over 60,000 generations. *Nature* 551:45–50.
18. Kryazhimskiy S, Plotkin JB. 2008. The Population Genetics of dN/dS . *PLoS Genet* 4:e1000304.
19. Bailey SF, Hinz A, Kassen R. 2014. Adaptive synonymous mutations in an experimentally evolved *Pseudomonas fluorescens* population. *Nature Communications* 2014 5:1 5:1–7.
20. Plotkin JB, Kudla G. 2010. Synonymous but not the same: the causes and consequences of codon bias. *Nature Reviews Genetics* 2011 12:1 12:32–42.
21. Tenaillon O, Barrick JE, Ribeck N, Deatherage DE, Blanchard JL, Dasgupta A, Wu GC, Wielgoss S, Cruveiller S, Médigue C, Schneider D, Lenski RE. 2016. Tempo and mode of genome evolution in a 50,000-generation experiment. *Nature* 536:165–170.
22. Cooper VS, Honsa E, Rowe H, Deitrick C, Iverson AR, Whittall JJ, Neville SL, McDevitt CA, Kietzman C, Rosch JW. 2020. Experimental Evolution In Vivo To Identify Selective Pressures during *Pneumococcal Colonization*. *mSystems* 5:e00352-20.
23. Scribner MR, Santos-Lopez A, Marshall CW, Deitrick C, Cooper VS, Hogan DA. 2020. Parallel evolution of tobramycin resistance across species and environments. *mBio* 11:e00932-20.

August 11, 2023

Prof. Ákos T Kovács
Universiteit Leiden
Institute of Biology Leiden
Sylviusweg
Leiden 2333BE
Netherlands

Re: mSystems00548-23R1 (Species and condition shape the mutational spectrum in experimentally evolved biofilms)

Dear Prof. Ákos T Kovács:

Thank you for revising your manuscript based on the second round of reviewer comments. I am pleased to inform you that your manuscript has been accepted, and I am forwarding it to the ASM Journals Department for publication.

Before it can be scheduled for publication, your manuscript will be checked by the mSystems production staff to make sure that all elements meet the technical requirements for publication. They will contact you if anything needs to be revised before copyediting and production can begin. Otherwise, you will be notified when your proofs are ready to be viewed.

If you would like to submit a potential Featured Image, please email a file and a short legend to msystems@asmusa.org. Please note that we can only consider images that (i) the authors created or own and (ii) have not been previously published. By submitting, you agree that the image can be used under the same terms as the published article. File requirements: square dimensions (4" x 4"), 300 dpi resolution, RGB colorspace, TIF file format.

We recognize that the video files can become quite large, and so to avoid quality loss ASM suggests sending the video file via <https://www.wetransfer.com/>. When you have a final version of the video and the still ready to share, please send it to mSystems staff at msystems@asmusa.org.

Sincerely,

Benjamin Wolfe
Senior Editor, mSystems

Journals Department
E-mail: mSystems@asmusa.org